# In Silico Analyses Indicate a Lower Potency for Dimerization of TLR4/MD-2 as the Reason for the Lower Pathogenicity of Omicron Compared to Wild-Type Virus and Earlier SARS-CoV-2 Variants

**DOI:** 10.3390/ijms25105451

**Published:** 2024-05-17

**Authors:** Ralf Kircheis

**Affiliations:** Syntacoll GmbH, 93342 Saal an der Donau, Germany; rkircheis@syntacoll.de; Tel.: +49-151-167-90606

**Keywords:** toll-like receptor (TLR), myeloid differentiation factor-2 (MD-2), TLR4/MD-2, Omicron, spike protein, SARS-CoV-2, COVID-19, cytokine storm, NF-kappa B

## Abstract

The SARS-CoV-2 Omicron variants have replaced all earlier variants, due to increased infectivity and effective evasion from infection- and vaccination-induced neutralizing antibodies. Compared to earlier variants of concern (VoCs), the Omicron variants show high TMPRSS2-independent replication in the upper airway organs, but lower replication in the lungs and lower mortality rates. The shift in cellular tropism and towards lower pathogenicity of Omicron was hypothesized to correlate with a lower toll-like receptor (TLR) activation, although the underlying molecular mechanisms remained undefined. In silico analyses presented here indicate that the Omicron spike protein has a lower potency to induce dimerization of TLR4/MD-2 compared to wild type virus despite a comparable binding activity to TLR4. A model illustrating the molecular consequences of the different potencies of the Omicron spike protein vs. wild-type spike protein for TLR4 activation is presented. Further analyses indicate a clear tendency for decreasing TLR4 dimerization potential during SARS-CoV-2 evolution via Alpha to Gamma to Delta to Omicron variants.

## 1. Introduction

After the emergence of the SARS-CoV-2 Omicron variant of concern (VoC) B.1.1.529 at the End of 2021 [1,2], the Omicron successor variants have successfully replaced all other variants worldwide. The appearance of Omicron has been associated with a dramatically increased infectivity, and broad evasion from therapeutic antibodies and neutralizing antibodies after vaccination or earlier infections with the virus, due to multiple mutations, particularly in the N-terminal domain (NTD) and receptor binding domain (RBD) of the spike protein [3,4,5,6,7,8].

Besides their higher infectivity and evasion from antibody response, all Omicron variants show a dramatic shift in cellular tropism with a lower virus replication in the lungs but faster replication in the upper respiratory tract [4,8,9]. This correlates with Omicron’s higher transmission rate within the population while causing less frequently acute respiratory distress syndrome (ARDS) and the severe clinical symptoms of COVID-19 [10]. The underlying molecular mechanisms for these reciprocal changes in cellular tropism with the appearance of the Omicron variants are not completely defined. A lower activation of Toll-like receptors (TLR) for Omicron variant has been hypothesized to be due to a changed charge distribution due to amino acid substitutions [11]. TLR-2 and TLR4 activation has been shown to be critically involved in SARS-CoV-2 induced pathogenicity [12,13,14,15,16,17,18], and binding of SARS-CoV-2 spike protein to TLR4 has been demonstrated for the wild-type virus [19]. However, so far no data regarding the dimerization of the TLR4/MD-2 have been shown. Notably, for the archetypal TLR4 ligand, i.e., LPS, dimerization of the TLR4/MD-2 has been demonstrated as the essential step for further downstream signaling [20]. Here, we show in silico analyses which indicate that (a) the spike protein trimer of SARS-CoV-2 can not only bind but also can result in dimerization of human TLR4/MD-2, dependent on the mode of binding, and (b) while the level of binding to TLR4 seems to be comparable to that of the SARS-CoV-2 wild-type, the Omicron spike protein shows a lower potency in inducing dimerization of TLR4/MD-2.

### 1.1. SARS-CoV-2 and the Pathophysiological Mechanisms of COVID-19

SARS-CoV-2—an enveloped positive-sense, single-stranded RNA virus—binds by its spike (S) protein to the angiotensin-converting enzyme-related carboxypeptidase-2 (ACE-2) receptor expressed on the surface of the target cells [21]. Besides, ACE2 alternative cellular receptors have been indicated as additional potential binding targets for the SARS-CoV-2, such as integrins, sialic acid, heparane sulfate, CD147, and neutropilin-1 [22,23]. The virus can enter the cell via two different routes, i.e., the highly efficient plasma membrane route or the cathepsin L-dependent endosomal entry route, dependent on whether TMPRSS2 (transmembrane serine protease 2) is co-expressed with ACE2 on the host cell membrane or not. While lung alveolar cells highly express TMPRSS2 [21], innate immune cells show no or only minimal expression of TMPRSS2; instead they use primarily the cathepsin L-dependent endosomal entry route [24,25,26].

The S protein of SARS-CoV-2—a class I viral membrane fusion protein—is present as a trimer, composed of a large ectodomain, a transmembrane anchor, and a C-terminal intracellular tail. The ectodomain comprises the receptor binding S1 subunit and the S2 subunit, which triggers membrane fusion. The receptor binding domain of S1 binds to ACE-2 on the target cells with high affinity, and S2 is responsible for fusion between the viral and host cell membranes. The S2 subunit consists of two heptad repeat (HR) regions (HR1 and HR2), the proteolytic site for the TMPRSS2 serine protease, a fusion peptide (FP), and the transmembrane (TM) domain. Between the S1 and S2 subunits, there is a SARS-CoV-2-unique poly-basic PRRAR furin-like cleavage site [27,28].

The binding receptor for SARS-CoV-2, ACE-2, is widely expressed among pulmonary and cardiovascular tissues. This explains the broad spectrum of pulmonary and extrapulmonary symptoms of SARS-CoV-2 infection, including cardiac and gastrointestinal manifestations [21,29,30]. While all Omicron variants generally show less clinical severity compared to the original Wuhan strain or earlier VoCs, severe cases of COVID-19 show a clinical picture that is similar to that seen in SARS-CoV-1- and MERS-CoV-infected patients [31,32,33,34]. While younger individuals show in most cases mild-to-moderate clinical symptoms, elderly individuals more often show a severe or critical clinical picture, and pre-existing comorbidities, including diabetes, respiratory and cardiovascular diseases, renal failure, sepsis, and male sex, are with higher probability associated with critical disease and mortality [34,35,36,37].

The pathophysiological manifestations of critical or fatal COVID-19 include a diffuse alveolar disease with capillary congestion, interstitial edema, platelet-fibrin thrombi, cell necrosis, and infiltrates of macrophages and lymphocytes [38]. Furthermore, severe signs of endotheliitis in various organs (including lungs, heart, kidney, and intestine) have been found as a direct consequence of viral involvement and of the host inflammatory response [29,30].

The molecular and cellular mechanisms of COVID-19 include (i) virus-induced cytopathic effects, (ii) viral evasion from the host immune response correlating with a dysregulated host IFN type I response [39], and (iii) highly dysregulated exuberant inflammatory and immune responses which correlate with the severity of disease and lethality [32,33,34]. Upregulated release of cytokines and chemokines, i.e., “cytokine storm”, plays a central role in the severity and lethality of SARS-CoV-2 infections [40,41,42,43,44,45]. The cytokine storm shows predominantly a Th1 pattern correlating with a hyperactivation of the NF-κB pathway as one of the critical signaling pathways for SARS-CoV-2 infection-induced proinflammatory cytokine/chemokine response, interconnected with other pathways, such as IL-6/STAT pathway [46,47,48,49,50,51,52]. Excessive activation of exuberant inflammatory responses involving endothelial cells, epithelial cells, and immune cells [44,45] can finally entail disturbances of the complement system, coagulopathies, disturbances of the bradikinine systems, finally triggering positive signaling feedback loops which accelerate inflammatory processes [53,54,55,56,57,58,59]. Additionally, vascular occlusion by neutrophil extracellular traps (NETs) [60,61] and coagulopathies, including thromboses and multiple micro-thromboses, are also hallmarks of COVID-19 disease [62,63,64,65].

### 1.2. Molecular Changes in the Spike Protein Omicron and Other VoCs

The Omicron variants have gathered multiple amino acid substitutions and deletions, particularly in the spike protein. The general structure of the spike protein and the mutations found in the Omicron variants and earlier VoCs are shown in Figure 1.

The highest number of the mutations in the Omicron spike protein are concentrated in the NTD, the RBD, and near to the S1/S2 cleavage site [3,4,5,6,7,67,68,69]. Two out of the three amino acid substitutions in the S1/S2 cleavage site region, i.e., P681H and H655Y, are found also in earlier VoCs, i.e., alpha (B.1.1.7) and gamma (P.1) variant, respectively (see Figure 1). In contrast, the S2 subunit is rather conserved, with few amino acid substitutions in the FP and HR1 regions and with only four unique mutations, which are common to all Omicron variants, i.e., N764K, D796Y, Q954H, and N969K. Notably, these are exchanges towards positively charged amino acids in three out of these four amino acid substitutions, and one loss of one negative charge (D796Y) [11].

### 1.3. Lower Pathogenicity of Omicron Compared to Previous VoCs

There is increasing evidence that Omicron, despite a significantly higher transmissibility and infectivity, shows a significantly lower rate of severe clinical courses compared to previous VoCs, including the Delta variant, i.e., the last VoC before Omicron [70]. Early reports on the lower number of severe clinical outcomes in Africa [1,2] are concordant with published studies from other geographical areas on the Omicron pandemic, such as the UK and USA, and have been confirmed by recent reports covering recent Omicron variants, including BA1 (B.1.1.529), BA2, BA4, BA5, XBB1.5 and EG5.1—all showing higher transmissibility but significantly lower pathogenicity [71,72,73,74,75,76,77,78,79,80].

The different behaviour of Omicron compared to early viral variants is well illustrated by plotting the ratios of deaths per million against cases per million over time. The resulting continuous line with changing colors (from violet to dark red), visualizing the different waves of infection in the COVID-19 pandemic over time, shows high death rates during earlier phases (steep circles), and a completely different, highly flattening plot for Omicron [81] (Figure 2).

Beside the significantly lower pathogenicity of Omicron, recent studies have indicated a generally narrowing cytokine pattern during evolution of the SARS-CoV-2 from the original Wuhan strain via the various VoCs, e.g., Alpha–Gamma–Delta–Omicron. Comparing the cytokine profiles of patients infected with different VoCs with those infected with the wild-type Wuhan variant, the studies showed that hypercytokinemia and cytokine storm become less threatening with the emergence of new mutations in the viral genome, this tendency being largely visible already with the Delta variant and with the most pronounced change for the Omicron variant [82,83].

The attenuated clinical picture of Omicron is further supported by experimental animal studies, which show equal or higher, mostly TMPRSS2-independent, viral replication in the upper respiratory tract, but lower pathogenicity in animal models. Studies on B.1.1.529 Omicron isolates regarding infection and disease in immunocompetent, human ACE2 expressing mice showed attenuated infection compared to earlier SARS-CoV-2 variants, with lower weight loss and lower viral burden in the lungs [8]. A milder pathological picture was also shown for Omicron infected Syrian golden hamsters compared to infection with earlier VoCs, i.e., Gamma and Delta or ancestral strain 614 G. Omicron infection caused only mild rhinitis and only minimal lesions in the lung. Viral antigens, RNA, and infectious virus titers were lower in the lungs of Omicron infected hamsters [84]. Furthermore, replication of Omicron was shown to be attenuated in human immortalized lung and intestinal cell lines, due to Omicron’s inefficient use of TMPRSS2 compared with wild-type SARS-CoV-2 and other VOCs. Omicron showed a significantly lower viral replication in both the upper and lower respiratory tracts of infected K18-hACE2 mice compared to mice infected with the wild-type strain or Delta variant and showed ameliorated lung pathology. In comparison to wild-type SARS-CoV-2 and various VoCs, including Alpha (B.1.1.7), Beta (B. 1.351) and Delta (B.1.617.2) variants, infection by Omicron caused significantly less reduction in body weight and the lowest mortality rate [85]. Furthermore, viral infection course, tissue tropism, and pathogenicity of SARS-CoV-2 D614G (B.1), Delta (B.1.617.2), and Omicron BA.1.1 (B.1.1.529) variants were studied in a feline model. While D614G- and Delta-inoculated cats showed clear signs of lethargy and increase in body temperatures, Omicron-inoculated cats remained subclinical and even showed an increase in weight. Intranasal inoculation of cats with D614G- and Delta variants resulted in high infectious virus shedding in nasal secretions, whereas markedly lower virus shedding was found in Omicron-inoculated animals. The tissue distribution of the Omicron variant was also largely changed in comparison to the D614G and Delta variants, showing lower viral loads in the respiratory tract. Importantly, while D614G- and Delta-inoculated cats showed clear signs of pneumonia, histology of the lungs of Omicron-infected cats showed only mild to modest inflammation [86].

Overall, these studies using different Omicron isolates have demonstrated a shift in cellular tropism with attenuated lung disease in rodents and cats, supporting human clinical data [1,2,71,72,73,74,75,76,77,78,79,80]. Still, the molecular mechanisms responsible for this reciprocal change in tropism in the Omicron variant are not completely defined so far. 

### 1.4. SARS-CoV-2 Activates Innate PRRs

Toll-like receptors (TLRs) are essential for recognition and elimination of pathogen-associated molecular patterns (PAMPs) from bacteria, viruses, and of self-derived damage-associated molecular patterns (DAMPs) released from dying or lytic cells. Typical PAMPs are nucleic acids, e.g., viral RNA and DNA, but also various glycoproteins, lipoproteins, and membrane components. 

Severity and disease progression in patients with COVID-19 was shown to correlate with the expression of MyD88, i.e., a key adaptor molecule in TLR-signaling, and the expression of TLR1, TLR2, TLR4, TLR5, TLR8 and TLR9 (but not TLR3) [12]. In this context, however, the expression levels of different TLR on the relevant target cells will be of importance. Highest expression levels are found for TLR4/MD-2 and TLR2 on innate immune cells, e.g., macrophages, and endothelial cells, whereas ~10 fold lower expression levels are found for TLR1, 6, and 7. In contrast to macrophages and endothelial cells, alveolar cell (in particular type 1) show generally very low levels of TLRs [11], suggesting that, primarily, the activation of TLR4 and TLR2 on macrophages and endothelial cells may be responsible for the excessive disturbances during severe and critical COVID-19 stages.

The question as to which of the TLRs have a particular high impact to the patho-physiological manifestations in severe and critical COVID-19 has been addressed in several studies. Khan et al. studied the inflammatory potency of different structural proteins of SARS-CoV-2 and showed that the spike protein strongly induces inflammatory cytokines and chemokines, such as IL-6, IL-1β, TNFα, CXCL1, CXCL2, and CCL2, but not IFNs in human and mouse macrophages. The spike protein was shown to activate the NF-κB pathway in a MyD88-dependent manner. Activation of the NF-κB pathway was abrogated in TLR2-deficient macrophages. Consistently, administration of the spike protein triggered the release of IL-6, TNFα, and IL-1β in wild-type mice but not in TLR2-deficient mice. Both, S1 and S2 subunits induced high NF-κB activation, with a higher potency found for S2 when compared on an equimolar basis [87].

Various studies have addressed the role of TLR4 in the pathogenicity of COVID-19. TLR4 is known to recognize multiple PAMPs derived from bacteria, viruses, and other pathogens [20]. Furthermore, TLR4 has been found to recognize certain damage-associated molecular patterns (DAMPs), e.g., high mobility group box 1 (HMGB1) and heat shock proteins (HSPs), which are released from dying or lytic cells during host tissue injury or viral infection [15,88]. TLR4 is mainly expressed on immune cells, in particular on macrophages and dendritic cells, but also on endothelial cells. Activation of TLR4 by PAMPs or DAMPs leads to the production of proinflammatory cytokines and/or the production of type I interferons and anti-inflammatory cytokines [15,88,89,90]. In contrast to other TLRs, TLR4 is present on both, the cell surface, where it recognizes, e.g., viral proteins before they enter the cell, but also in endosomes [20,91]. As demonstrated in detail with the archetypal TLR4 agonist from gram-negative bacteria, i.e., lipopolysaccharide LPS, dimerization of the TLR4/MD-2 complex represents the central molecular step within the TLR4 activation cascade, with hydrophobic chains and charged groups of the ligand essential for binding and dimerization of TLR4/MD2 [91,92,93,94,95].

The impact of TLR4 in the pathogenies of COVID-19 has been demonstrated in multiple studies [14,15,16,17,18,19,96]. The release of IL1β triggered after infection by SARS-CoV-2 was shown to be abrogated by the TLR4-specific inhibitor Resatorvid. The SARS-CoV-2 spike trimer was demonstrated by surface plasmon resonance (SPR) to bind to TLR4 with an affinity of ~300 nM, which is comparable to that of many virus-receptor interactions. A monocytic cell line was treated with either the whole spike protein trimer, or the NTD or the RBD of spike protein, respectively. Only the trimeric protein, but not the NTD or RBD alone, induced IL1β and IL6, and this induction was blocked by the TLR4 inhibitor Resatorvid. The spike protein also induced IL1β in a murine macrophage cell line in a TLR4- and MyD88-dependent manner as well as in primary bone marrow-derived macrophages and peritoneal macrophages from wild-type but not in macrophages derived from TLR4-deficient mice. IL1β expression induced by spike protein was blocked by the NF-κB inhibitor JSH-23. Activation of TLR4 by spike protein was shown to be non-related to ACE2, TMPRSS2, or virus entry as shown on macrophages from ACE2-deficient mice or mice overexpressing human ACE2. Treatment with ACE2 inhibitor (MLN-4760) or soluble ACE2 or TMPRSS2-specific inhibitor (Bromhexine hydrochloride) also had no effect on the induction of IL1β by LPS or spike protein. Notably, the induction of IL1β by the spike protein trimer from SARS-CoV-2 or SARS-CoV was comparable to LPS treatment [17].

Exposure of peritoneal exudate-derived murine macrophages to SARS-CoV-2 spike protein S1 subunit induced TNF-α, IL-6, IL-1β, and nitric oxide, and activated NF-κB and c-Jun N-terminal kinase signaling pathways. Treatment with a TLR4 antagonist attenuated pro-inflammatory cytokine release and activation of intracellular signaling by S1 and lipopolysaccharide, suggesting that the S1 subunit activates TLR4 signaling in murine and human macrophages [96]. Furthermore, the SARS-CoV-2 spike S1 domain was shown to be a TLR4 agonist in rat and human cells inducing a pro-inflammatory M1 phenotype in human monocyte-derived macrophages. Adult rat cardiac tissue resident macrophage-derived fibrocytes were treated with either LPS or recombinant SARS-CoV-2 spike S1 glycoprotein and showed TLR4 activation and upregulation of ACE2. The effects of spike S1 and LPS could be blocked by the specific TLR4 inhibitor Resatorvid, confirming the spike S1 subunit as a TLR4 agonist. Confocal immunofluorescence microscopy showed a 1:1 stoichiometric co-localization of spike S1 with TLR4 in rat and human cells, these data confirmed by proximity ligation assays [18].

Moreover, previously published in silico studies have shown the strongest protein–protein interaction of the spike glycoprotein of SARS-CoV-2 with TLR4 compared to other TLRs [19]. Furthermore, a model has been proposed for binding of SARS-CoV-2 spike glycoprotein to TLR4 leading to the activation of TLR4 signaling characterized by increased cell surface expression of ACE2 that facilitates virus entry. Finally, SARS-CoV-2-induced myocarditis and multiple-organ damage was correlated to TLR4 activation and hyperinflammation in COVID-19 patients. All these data indicate that TLR4 may significantly contribute to the pathogenesis of SARS-CoV-2 indicating TLR4 as a promising therapeutic target in COVID-19, which is supported also by the fact that TLR4 antagonists have been previously used in sepsis and in other antiviral contexts [18].

## 2. Results

### 2.1. In Silico Analyses of TLR4/MD-2 Binding and Dimerization by Omicron Spike Protein Compared to Wild-Type Virus Spike Protein

Studies with the archetypal TLR4 agonist LPS and its derivatives have demonstrated that agonists share the functional feature of binding both TLR4 and the hydrophobic pocket of the myeloid differentiation factor 2 (MD-2, Lymphocyte antigen 96, Ly96) that is part of the TLR4/MD-2 complex. The high-affinity binding leads to dimerization of the TLR4/MD-2 complex. Dimerization of the TLR4-MD-2 complex is an absolute pre-requisite for TLR4 activation, leading to the recruitment of the intracellular adaptor protein MyD88 coupled to intracellular signaling [20,91,92,93,94,95].

We have analyzed in silico the binding of the spike protein trimer from wild-type SARS-CoV-2 (PDB 6ZGG) (PDB DOI: https://doi.org/10.2210/pdb6ZGG/pdb EM Map EMD-11205: EMDB EMDataResource, accessed on 6 May 2024) [97] to the human TLR4/MD-2 complex (PBD 3FXI) (PDB DOI: https://doi.org/10.2210/pdb3FXI/pdb, accessed on 6 May 2024) [98] and have compared this with binding of the spike protein trimer from Omicron variant (PDB 7TL9) (PDB DOI: https://doi.org/10.2210/pdb7TL9/pdb EM Map EMD-25984: EMDB EMDataResource, accessed on 6 May 2024) [99]. Importantly, for the human TLR4/MD-2, a PDB model based on dimerization of human TLR4/MD-2 by LPS (PDB 3FXI) [98] was chosen in order to analyze for simultaneous binding of the spike protein trimers to two TLR4/MD-2 complexes. Furthermore, for both, wild-type virus, and Omicron, PDB spike protein trimer models with both, monomers in closed (“down”) AND in open (“up”) state, were chosen to cover the broadest range of configurations of the spike protein trimer.

We have analyzed the potency for dimerization of the human TLR4/MD-2 complex (PDB 3FXI) which will require interaction with two TLR4/MD-2 partners (i.e., TLR4/MD-2 (I) and (II), respectively). Docking studies using HDOCK software (©Lab of Biophysics and Molecular Modeling, huanglab@hust.edu.cn) [100] and PDB driven data showed that both the spike protein trimer from wild-type virus (PDB 6ZGG) as well as from Omicron variant (PDB 7TL9) show a comparable binding to the human TLR4/MD-2 complex with docking scores for the top 10 models ranging between −348 and −290 and −368 and −297, respectively (Table 1 and Table 2).

However, differences were found when comparing the actual site of binding of the spike protein trimer to the human TLR4/MD-2 complex and regarding the question which of the subunits of the spike protein are involved. For the wild-type SARS-CoV-2 four out of the ten (4/10) top binding models (models 1, 3, 8, 10) (Figure 3A,B) showed no dimerization, whereas two models (2/10) showed dimerization of TLR4/MD-2 induced by the S1 subunit of the spike protein (models 4, 5) (Figure 3C,D). Interestingly, four out of ten (4/10) models showed dimerization of TLR4/MD-2 complex by interaction with both, S1 and S2 subunits (models 2, 6, 7, 9) (Figure 3E–H). In contrast, for the Omicron spike protein, seven out of ten (7/10) top binding models showed no dimerization of TLR4/MD-2 (models 2, 3, 4, 5, 7, 8, 9) (Figure 4A–D), two out of ten (2/10) showed S1 induced dimerization (models 1, 10) (Figure 4E,F), but only one model (1/10) showed TLR4/MD-2 dimerization induced by S1 and S2 subunit (model 6) (Figure 4G).

To interpret the differences in dimerization potency vs. binding activity to TLR4/MD-2 between wild-type and Omicron variant calculated for various models and their biological consequences, the following aspects have to be taken into consideration: as the TLR4/MD-2 recognizes a broad variety of danger signals, including large viral proteins [15,20,89,90,91,92] that in most cases do not fit spatially into the hydrophobic pocket of MD-2 as used by the relatively small archetypal TLR4 ligand LPS [20,98], alternative binding modalities of recognized danger signals to the TLR4 lycine rich rings and/or the MD-2 have to be assumed. The functional similarity of the various TLR4/MD-2 ligands can therefore not be a structural mimicry of the exact dimensions and configurations of LPS but rather the shared potential to bind both TLR4 and/or MD-2 of the two dimerization partners, i.e., TLR4/MD-2 complexes I and II, resulting in dimerization. It is therefore conceivable that, for the rather large spike protein trimer of wild-type SARS-CoV-2 or Omicron, respectively, there is not only one (i.e., that with the highest binding affinity) binding modus, but rather the sum of all high affinity configurations, which are able to induce dimerization of the TLR4/MD-2 complex. In this context, the higher probability for multiple binding configurations of the wild-type spike protein to the TLR4/MD-2 dimer (I and II) complexes compared to Omicron spike protein can be expected to correlate with the difference in the amplitude of general TLR4 activation in the relevant cells.

Conclusively, these data indicate, that while binding of the spike protein trimers to TLR4/MD-2 seem to be comparable, the mode of binding with respect to dimerization is different for the Omicron variant compared to the wild-type virus. The differences between the virus variants were most striking for those binding configurations which use interaction of both subunits, S1 and S2, with the TLR4/MD-2. Only one configuration with S1 and S2 binding was calculated for the Omicron spike protein trimer compared to four configurations for the wild-type virus. Three configuration found for the wild-type spike protein trimer seem to be lost due to amino acid substitutions in the Omicron variant, including two very similar configurations of binding of the wild-type spike protein with TLR4/MD-2, i.e., models 2 and 6 (see Figure 3E,F).

Performing analogous in silico modelling on other VoCs, including Alpha, Gamma, and Delta variant, showed a decrease in the number of dimerization modes after the Alpha variant, with 6, 7, 5, 4, and 3 modes of TLR4/MD2 dimerization for the wild-type, Alpha, Gamma, Delta, and Omicron VoCs, respectively (Figures for Alpha, Gamma, and Delta variants in Appendix A). These data correlate with data from recent studies, which indicate a narrowing cytokine pattern during evolution of the SARS-CoV-2 [82,83].

Analyzing in detail the binding and dimerization regarding the amino acid interactions between the spike trimer of wild-type SARS-CoV-2 with TLR4/MD-2 in model 2, the salt interaction, polar, and non-polar interaction of the spike trimer with TLR/MD-2 dimerization partner I and with the TLR4 part of dimerization partner II, respectively, were calculated. Figure 5. shows the multiple interactions of the S protein trimer with the dimerization partner TLR/MD-2 (I) and significantly fewer interactions with the second dimerization partner, i.e., TLR4 (II), indicating binding to TLR4 (II) as being potentially limiting for dimerization. Notably, ionic interactions with both partners were also identified, e.g., Lys795 of the S protein with Glu563 on TLR4 (II), together with multiple polar interactions S2 region around Lys921, Asn925, Asn928, and Asp936 with TLR4 (II).

Figure 6A,B shows models 2 and 6, respectively, for binding and dimerization of TLR4/MD-2 (I) and TLR4 (II) by the wild-type virus S protein trimer, showing binding of the S1 subunit to TLR4 (I) and MD-2 (I), and S2 binding to TLR4 (II), without involvement of MD-2 (II). Figure 6C shows at a higher zoom the interaction of S protein trimer with TLR4 (I), MD-2 (I) and TLR4 (II), respectively, with the main ionic and polar interacting amino acids indicated (blue—positively charged amino acids, red—negatively charged amino acids, pink—polar amino acids), dotted lines—ionic interactions.

Dimerization of TLR4/MD-2 is supported by binding of the spike protein S1 subunit (mainly NTD) to TLR4 (I) and MD-2 (I) and binding of S2 subunit (FP and HR1 region) to TLR4 (II). Six salt bridges and multiple polar and non-polar interactions are involved in the binding (see Figure 5 and Figure 6).

When the 3D structures of the monomers of the spike proteins of wild type (PDB 6ZGG) and Omicron (PDB 7TL9) are aligned a higher degree of alignment is seen for the S2 subunits compared to the S1 subunit (Figure 7A) except for a region around the Fusion peptide, which seems to be slightly shifted in the Omicron S protein compared to the wild-type S protein (Figure 7B).

And this region seems to be involved in the interaction of the spike protein with TLR4 (II), with salt bridge between Lys795 of the spike protein and Glu563 of TLR4 and multiple polar interactions, including Asp796 (spike protein) and Gln565 (TLR4). This interaction, however, is only found for the wild-type spike protein trimer (PDB 6ZGG) in HDOCK docking studies models 2 and 6 (Figure 7C). In contrast, in the Omicron spike protein, Asp796 is exchanged by Tyr796 (Figure 7B). Importantly, in this region no high affinity interaction between the Omicron spike protein and with TLR4 (II) are found by docking studies using HDOCK among the top binding models, indicating that the spatial shift in FP region (also previously noted by Gobeil et al. [99]) together with the amino acid exchange of Asp796 by Tyr796 are responsible for abrogating an efficient binding to TLR4 in this region.

Conclusively, the amino acid exchanges in the Omicron spike protein in the FP / HR1 region of the S2 subunit are likely to lead to a lower potency in supporting dimerization of the TLR4/MD-2 complex compared to wild-type. Additionally, the docking studies with the other VoCs (Appendix A) indicate that also amino acid substitutions present in earlier VoCs, e.g., Delta variant, may contribute to a decreased potential of the spike protein for TLR4 dimerization.

### 2.2. Binding and Dimerization of TLR2/TLR1 and TLR2/TLR6 Heterodimers

Given the high expression of both TLR4 and TLR2 in innate immune cells, the binding and dimerization of TLR2 heterodimers (with TLR1 and TLR6, respectively) have been studied (data shown in Appendix A). There was a moderate difference between spike protein of Omicron variant vs. spike protein of wild-type virus regarding the number of dimerization modes, with 8 vs. 7 and 4 vs. 3 dimerization modes, respectively. 

In this context, it should be noted that the cellular expression of TLR4/MD-2 complexes is several-fold higher than the expression of TLR1 and TLR6 (see new Figure 8B,C), which serve as dimerization partners for TLR2 heterodimers.

### 2.3. Mechanistic Model for Lower Pathogenicity of Omicron Variants Compared to Wild-Type Virus and Early VoCs

The in silico analyses presented in this study have shown that, despite a comparable binding activity, the Omicron spike protein shows a lower potency for dimerization of human TLR4/MD-2, due to amino acid substitutions and slight conformational shifts in the FP/HR1 region of the S2 subunit [99]. TLR4/MD-2 is preferably expressed on innate immune cells, such as macrophages and dendritic cells, and—to some lower extent—endothelial cells [11]. These cells may be largely responsible for excessive immune stimulation, including cytokine storm found in severe and critical cases of COVID-19 [11,44,101]. Notably, this TLR-mediated immune hyperstimulation is to some extent uncoupled from virus replication, which is largely occurring in ACE2-expressing epithelial cells (such as lung alveolar cells, cells of the intestinal tract etc.) and which depends on proteolytic cleavage of the spike protein by furin. Viral uptake is accelerated by TMPRSS2 co-expression enabling uptake via the highly efficient plasma membrane route. In contrast, in TMPRSS2 negative innate immune cells, such as macrophages, the virus is taken up preferably by clathrin-coated pits and directed into endosomes, where the proteolytic cleavage is taken over by endosomal enzymes, such as cathepsin L [24,25,26].

Notably, TLR4/MD-2 is the only TLR which is found on the cell surface as well as in the endosomes. Binding and dimerization of TLR4/MD-2 by the SARS-CoV-2 spike protein leads to downstream signaling, resulting in NF-κB pathway activation and cytokine release of TNFα, IL-1, IL-6, IL-8, IL-12, together with HIF-1α, chemokine release, upregulation of adhesion and cell activation molecules, and enhanced expression of furin [11,102,103]. The lower potency for TLR4/MD-2 dimerization of Omicron spike protein compared to wild-type virus or earlier VoCs will—despite comparable binding affinity—results in a lower TLR4-NF-κB—signaling and lower cytokine release (Figure 8).

## 3. Discussion

Omicron variants have successfully replaced all other variants worldwide. The emergence of the Omicron variants has been associated with a dramatically increased infectivity and broad evasion of humoral response induced by vaccination or earlier infections due to multiple mutations in the spike protein [3,4,5,6,7,8].

In parallel a shift in cellular tropism with a lower virus replication in the lungs but higher replication in the upper respiratory system [8,9], and a lower pathogenicity with less involvement of the lower respiratory system, have been found as general observations in clinical and preclinical studies. The molecular mechanisms responsible for these reciprocal changes in cellular tropism were until now not completely defined, and a lower activation of Toll-like receptors (TLR) for Omicron has been hypothesized based on changes in the charge distribution due to amino acid substitutions [11]. TLR-2 and TLR4 activation have been shown to play an essential role in SARS-CoV-2 induced pathogenicity [12,13,14,15,16,17,18], and binding of SARS-CoV-2 spike protein to TLR4 has been demonstrated for the wild-type virus [19] and recently also for the Omicron spike protein [104]. Notably, no dimerization of the TLR4/MD-2 have been demonstrated so far.

In the present in silico study, we show that (a) the spike protein trimer of SARS-CoV-2 can induce not only binding but also dimerization of human TLR4/MD-2, dependent on the mode of binding, and (b) while the binding affinities to TLR4 seems to be comparable to that of the SARS-CoV-2 wild-type, the Omicron spike protein was found to have a lower potency to induce dimerization of TLR4/MD-2. The in silico data from the present study indicate that there is a higher probability for multiple binding configurations, which induce dimerization of the TLR4/MD-2 complex by the wild-type spike protein compared to Omicron spike protein. This difference can be expected to lead to the difference in the amplitude of general TLR4 activation in the relevant cells, preferably innate immune cells, and endothelial cells. The dimerization of TLR4 activating the relevant downstream signaling pathways will lead to excessive gene expression for a broad range of pro-inflammatory cytokines and chemokines, adhesion molecules, and acute phase proteins. Furthermore, the highly activated NF-κB pathway, directly or via HIF-1α activation or via cytokine release (such as IL-12) is expected to stimulate furin expression in the cells. In this line, viral infections, cancer, hypoxia, HIF-1α and cytokines (e.g., IL-12) have been found to significantly enhance furin expression [11,102,103].

In silico modelling of TLR4 dimerization induced by other VoC, including Alpha, Gamma, and Delta variants, showed a decrease of the number of dimerization modes after the Alpha variant, with 6, 7, 5, 4, and 3 modes of TLR4/MD2 dimerization for the wild-type, Alpha, Gamma, Delta, and Omicron VoCs, respectively (see Appendix A). These data correlate well with reported cytokine release patterns compared in a recent study. Korobova et al. have shown a narrowing cytokine pattern during evolution of the SARS-CoV-2 from the original Wuhan strain via the various VoCs, e.g., Alpha- Delta- Omicron. The study compared the cytokine profiles of patients with different variants of SARS-CoV-2 and showed that hypercytokinemia and cytokine storm become less threatening with the emergence of new mutations in the viral genome, being already visible with the Delta variant and most pronounced for the Omicron variant [82]. A similar tendency of decreasing pathogenicity has been shown by Barh et al. showing decreasing pathogenicity (correlating with decreasing cytokine release) with virus evolution, with Wuhan strain > Gamma > Delta > Omicron [83].

The TLR binding and dimerization pattern under virus evolution from wild-type–Alpha–Gamma–Delta–Omicron may slightly differ for different TLRs, as shown in the present study for TLR2/TLR1 and TLR2/TLR6 heterodimers, which may contribute to a not completely homogenous tendency among different cytokines [82]. However, the multifold higher expression of TLR4/MD-2 compared to the other TLR species (see Figure 8B,C, and Ref. [105]) indicate that activation of TLR4/MD-2 can be expected to largely define the resulting net TLR activation.

It is the opinion of the author that, beside the world-wide vaccination campaign, the emergence of the Omicron variants with their higher infectivity and immune evasion potential outcompeting all earlier VoCs, in combination with the shift in cellular tropism and lower pathogenicity, is one of the two main reasons for successful overcoming of the pandemic. Understanding the underlaying molecular mechanisms will be essential not only for understanding SARS-CoV-2 and COVID-19 but also in preparation for controlling upcoming viral diseases in the future.

The correlation between TLR4 dimerization and activation with pathogenicity of the SARS-CoV-2 variants indicate TLR4 as a potential therapeutic target for treatment of COVID-19. While so far there are no FDA approved TLR4 antagonists available, intensive research [106] on TLR4 as a therapeutic target is ongoing, with emphasis on respiratory and neurological complications of SARS-CoV-2 [107], also with respect to accumulating data for long-term nervous system consequences after COVID-19 [108,109]. Beside various TLR4 antagonists which inhibit binding to or dimerization of TLR4, also inhibitors of the downstream signaling pathways, including inhibitors of the NF-κB pathway, are promising targets for therapeutic intervention [50,51].

Beside chemical TLR4 antagonists, antibody-based therapies targeting TLR4/MD-2 binding and/or dimerization may be highly interesting considering the high binding affinity and specificity, and the good safety profile shown for humanized monoclonal antibodies [110], or recent further developments, such as nanobodies [111]. Still, antibody-based therapies will face similar challenges encountered by previous agents, such as a continued evolution of SARS-CoV-2 and emergence of new escape mutations, in particular for the ACE2 binding RBD of the spike protein, as a major force for generation of new virus variants. Strategies to potentially mitigate this may include alternative target sites of the virus, preferably on the highly exposed spike protein. In this respect epitopes and regions involved in TLR4 binding and/or dimerization may be of particular high interest. While not interfering with the main receptor of SARS-CoV-2 for cellular uptake (thus not exposed to such an extremely high selection pressure as the ACE2 binding sites), these therapies could ameliorate the hyperactivation of TLR4 associated with the cytokine and chemokine release syndrome. In this direction, very encouraging data have been reported from a Phase 2 clinical study with an experimental monoclonal antibody (EB05, Edesa Biotech, Inc. Markham, Ontario L3R 5H6, Canada) in hospitalized COVID-19 patients. A 68% reduction of mortality risk compared to placebo plus standard of care (including dexamethasone, IL-6 inhibitors) at 28 days was reported for this monoclonal antibody which targets TLR4 and inhibits TLR4 signaling [112].

Limitations of the present study regard the focus on a limited panel of TLRs which are relevant for recognizing the spike protein of SARS-CoV-2. Beside spike protein recognition, recognition of the nucleic acid components of the virus by TLR3,7,8,9 and of other structural and non-structural viral proteins can be expected to have an impact on the overall TLR activation by SARS-CoV-2. Furthermore, different TLRs may make different contributions to anti-viral immune activation and to pathophysiological hyperstimulation as two sides of TLR activation [113]. Another limitation of the present study is that all modelling is based on in silico analyses using various data bases (PDB, HDOCK, Human Protein Atlas) in combination with reported experimental data from the literature, without showing direct experimental evidence. Further studies in cell culture and animal models as well as clinical data will be necessary to confirm or further develop the hypotheses raised in the present study. 

## 4. Materials and Methods

The sequences and 3D models of the spike protein trimer from wild-type SARS-CoV-2 (PDB 6ZGG) [97], Omicron variant (PDB 7TL9) [99], Alpha variant (PDB 7LWU) [114], Gamma variant (PDB 7M8K) [115], Delta variant (PDB 2W92) [116], and of the human TLR4/MD-2 complex (PDB 3FXI) [98], TLR2/TLR1 heterodimer (PDB 2Z7X) [117], TLR2/TLR6 heterodimer (PDB 2A79) [118], were derived from the RCSB PDB Protein Data Bank (https://www.rcsb.org/, accessed on December 2023 and January–May 2024). For all spike protein models only 1-RBD-up conformation models, i.e., with monomers in closed (“down”) AND in open (“up”) state, were chosen to cover the broadest range of configurations of the spike protein trimer.

In silico docking studies were performed using HDOCK docking software (http://hdock.phys.hust.edu.cn/) [100]. Alignments were performed using the RCSB PDB Protein Data Bank (https://www.rcsb.org/)

The expression profiles for relevant receptors on different target cells normalized to nTPM (i.e., Transcripts per million protein coding genes) have been derived from the Human Protein Atlas (https://swww.proteinatlas.org/) [119].

## 5. Conclusions

The recently appearing Omicron variant shows surprisingly reciprocal changes in cellular tropism of the upper and lower respiratory system, together with lower pathogenicity which cannot be explained simply by the changed binding to the ACE2 receptor or immune escape. Here, we present in silico analyses as part of a mechanistic model, which indicate that amino acids substitutions in the S2 subunit of the spike protein unique to the Omicron variants, as compared to previous VoCs, may disturb the recognition by innate Pattern-recognition receptors (PRRs), such as TLR4/MD-2, resulting in lower activation of downstream signaling pathways, and lower immune hyperactivation.

## Figures and Tables

**Figure 1 ijms-25-05451-f001:**
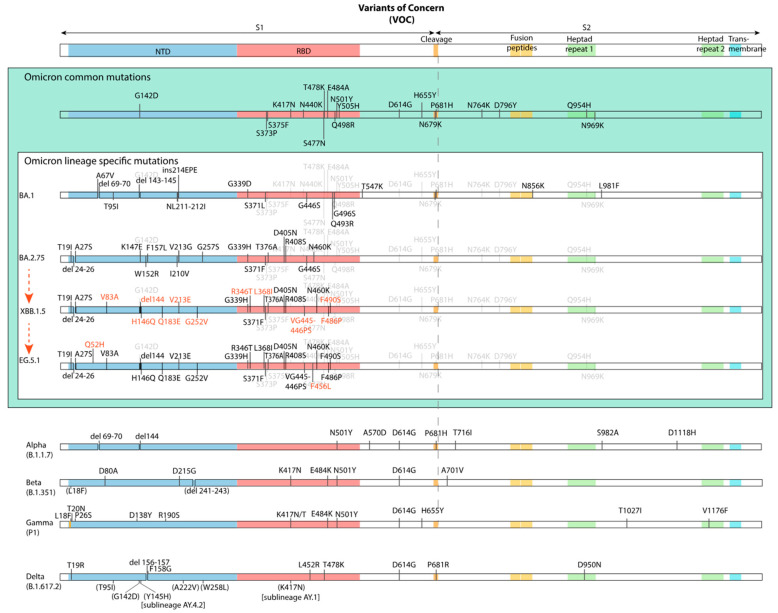
Mutations in the spike protein in Omicron variants compared to wild type SARS-CoV-2 and earlier VoCs. Mutations which are common to all Omicron variants are shown in the upper (green) part. Picture derived from [66].

**Figure 2 ijms-25-05451-f002:**
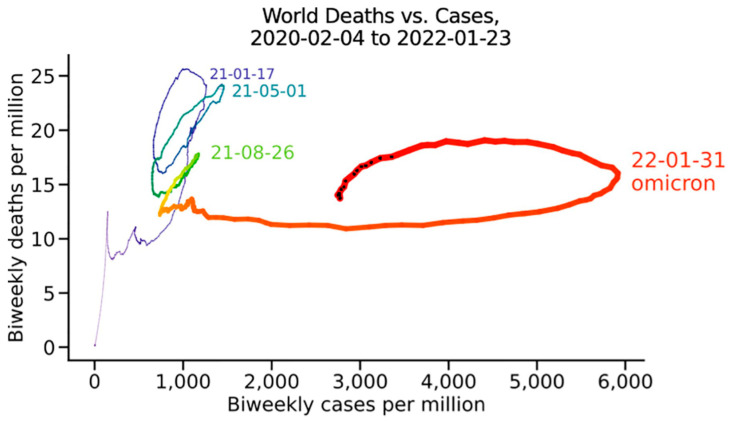
All reported cases and deaths worldwide, reproduced from Ref. [79].

**Figure 3 ijms-25-05451-f003:**
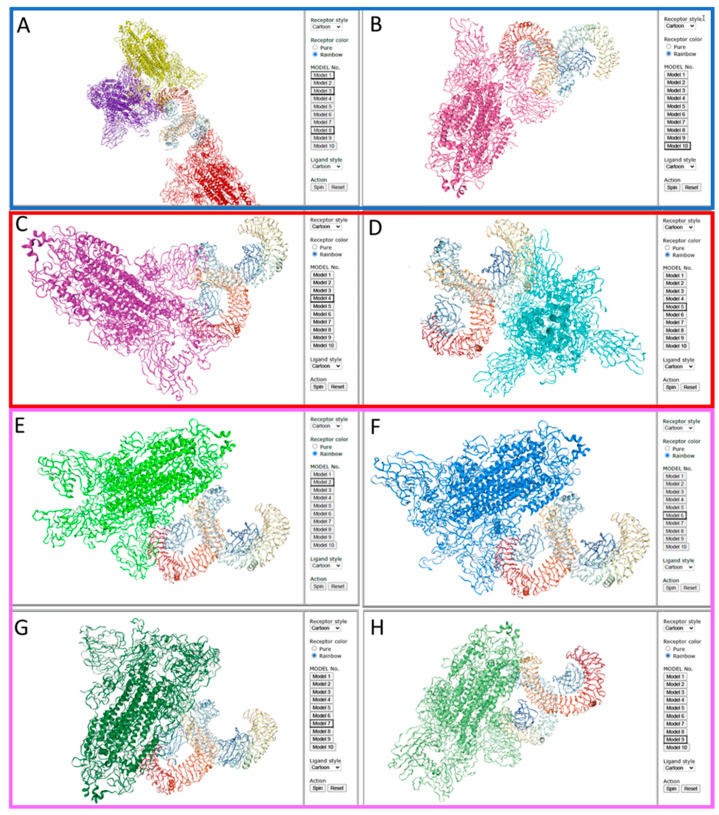
Top ten docking models for spike protein trimer from the wild-type SARS-CoV-2 (PDB 6ZGG) to the human TLR4/MD-2 complex (PBD 3FXI) calculated using HDOCK software (http://hdock.phys.hust.edu.cn/, accessed on 16 December 2023). Four out of the ten top binding models (models 1, 3, 8, 10) show binding to TLR4/MD-2 without dimerization ((**A**,**B**) in blue frame), two models (models 4, 5) show dimerization of TLR4/MD-2 induced by the S1 subunit of the spike protein ((**C**,**D**) in red frame), four out of ten models (models 2, 6, 7, 9) show dimerization induced by both, S1 and S2 subunits ((**E**–**H**) in pink frame).

**Figure 4 ijms-25-05451-f004:**
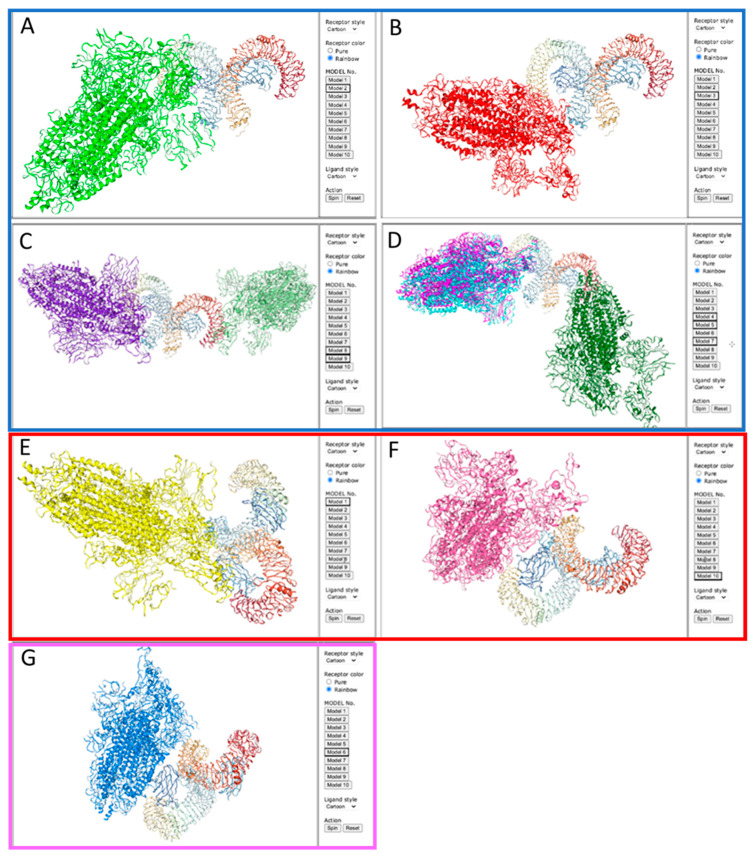
Top ten docking models for spike protein trimer from the Omicron SARS-CoV-2 (PDB 7TL9) to the human TLR4/MD-2 complex (PBD 3FXI) calculated using HDOCK software (http://hdock.phys.hust.edu.cn/, accessed on 16 December 2023). Seven out of ten (7/10) top binding models (models 2, 3, 4, 5, 7, 8, 9) show binding without dimerization of TLR4/MD-2 ((**A**–**D**) in blue frame), two out of ten showed S1 induced dimerization (models 1, 10) ((**E**,**F**) in red frame), one model (model 6) shows TLR4/MD-2 dimerization induced by S1 and S2 subunit ((**G**) in pink frame).

**Figure 5 ijms-25-05451-f005:**
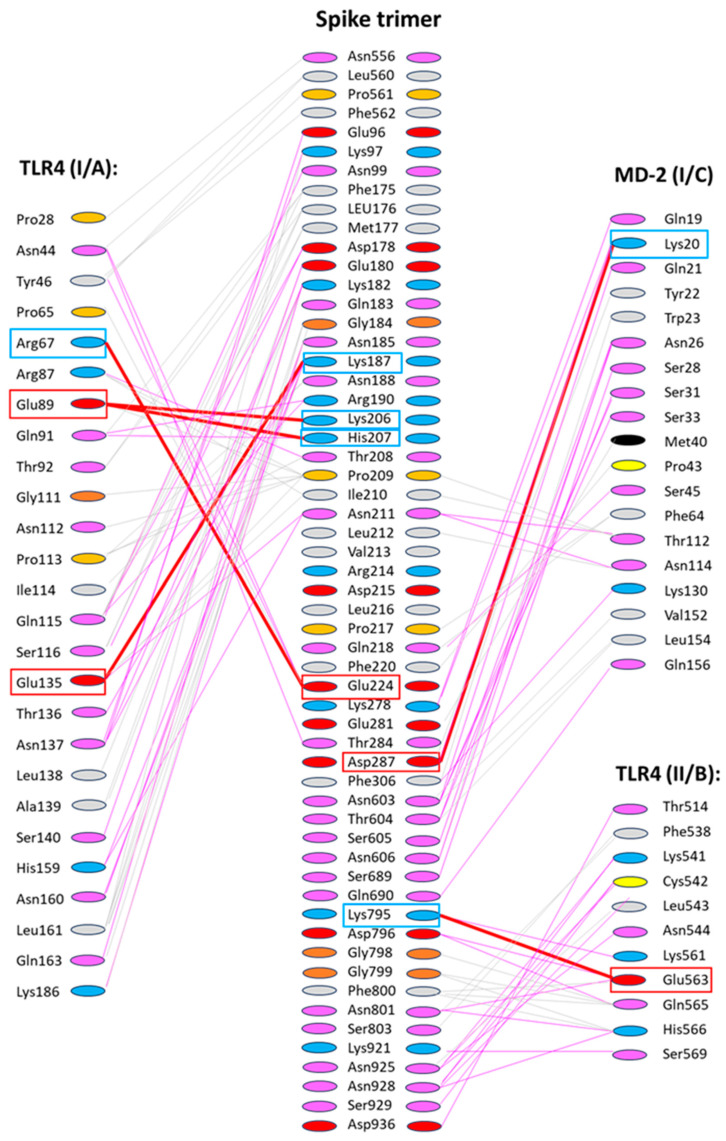
Salt interaction (thick red lines connecting two colored boxes which highlight involved amino acid residues), polar (pink lines), and non-polar (grey lines) interaction of the spike trimer with TLR/MD-2 dimerization partner I and with the TLR4 part of dimerization partner II as calculated by HDOCK software (http://hdock.phys.hust.edu.cn/, accessed on 16 December 2023) are shown.

**Figure 6 ijms-25-05451-f006:**
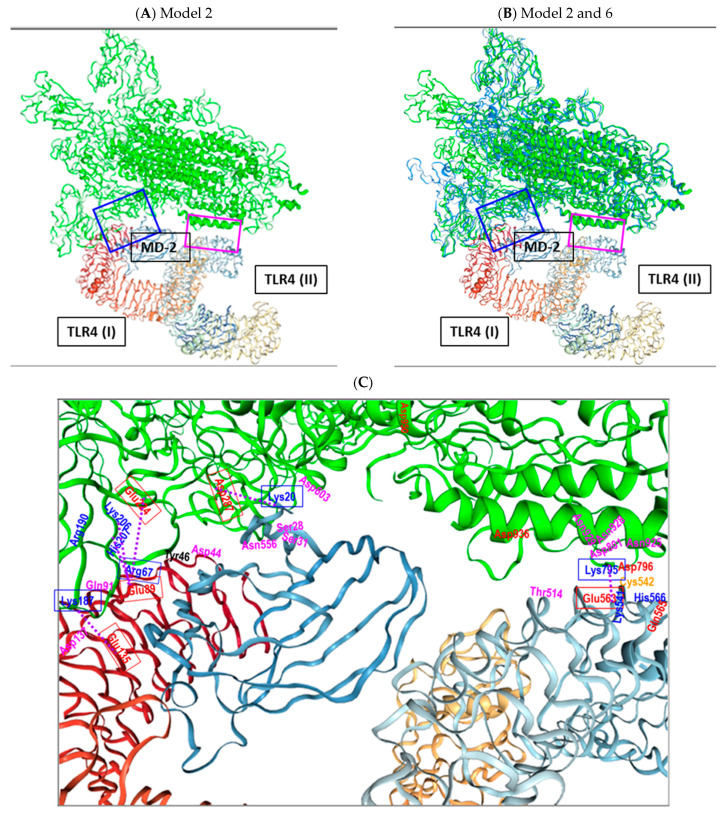
HDOCK generated model 2 (**A**) and models 2 and 6 superimposed (**B**) for binding and dimerization of TLR4/MD-2 (I) and TLR4 (II) by the wild-type virus S protein trimer (model 2—green; model 6—blue). Blue and pink frames show interaction areas with TLR4/MD-2 (I) and TLR4 (II), respectively (**C**) higher zoom of the interaction of S protein trimer (green color, upper part) with TLR4 (I) (spans from left side/red color to right side/yellow color)), MD-2 (I) (left side/center, blue color) and TLR4 (II) (right side, light blue color), colors of amino acids: blue—positively charged amino acids, red—negatively charged amino acids, pink—polar amino acids, orange – cysteine, dotted lines—ionic interactions.

**Figure 7 ijms-25-05451-f007:**
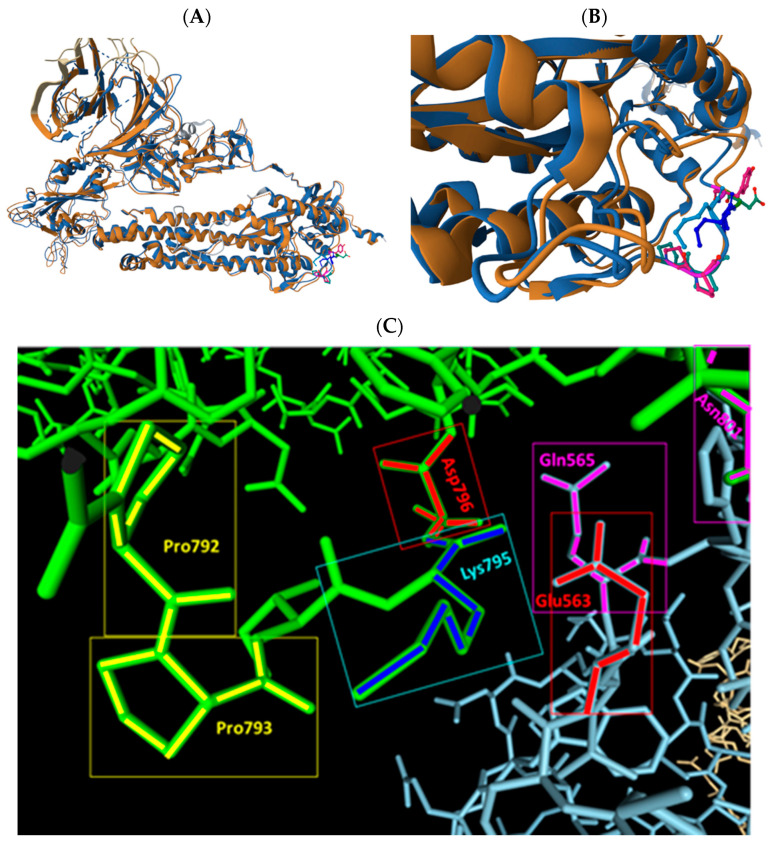
Pairwise alignment of monomers of S proteins of wild type (PDB 6ZGG) and Omicron (PDB 7TL9) (**A**), higher zoom (**B**). The two Lys795 are shown in dark blue vs. light blue and Asp796 and Tyr796 are shown in green and pink in the wild-type (6ZGG) vs. Omicron (7TL9) models, respectively. Pro792 and Pro793 are also shown for both models (**A**,**B**). (**C**) Spatial interaction between Lys795 (dark blue) and Asp796 (red) of wild-type virus spike protein with Glu563 (red) and Gln565 (pink) in TLR4 (II) were calculated and displayed using HDOCK software (http://hdock.phys.hust.edu.cn/, accessed on 9 February 2024). The two proximate prolines (Pro792 and Pro793) are shown in yellow.

**Figure 8 ijms-25-05451-f008:**
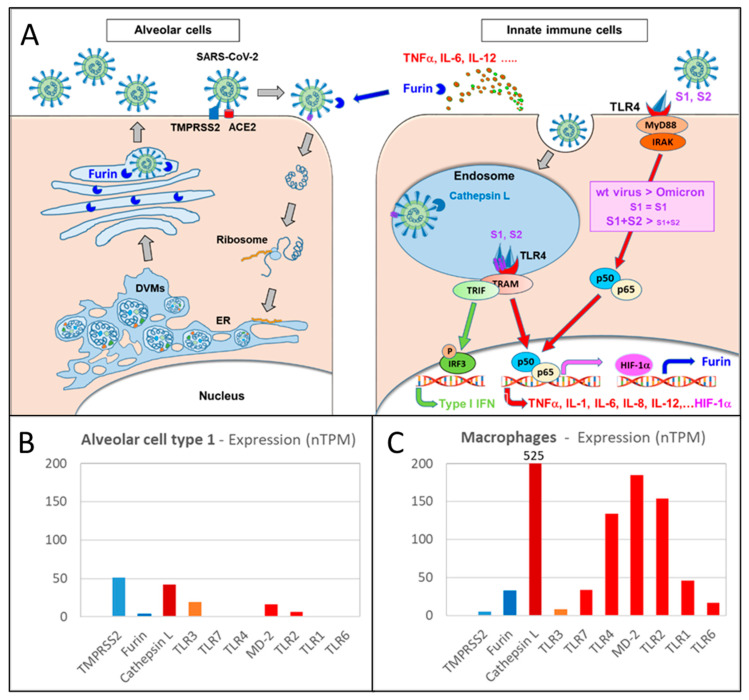
(**A**) The SARS-CoV-2 binds to ACE2 followed by proteolytic cleavage by TMPRRS2 and fusion with the host cell membrane followed by uptake of viral RNA into the host cell. Transcription of the viral RNA and translation of viral non-structural and structural proteins occurs in double-membrane vesicles (DMVs), which are formed after remodeling of the endo-plasmatic reticulum (ER). The newly produced viral components assemble into virus particles which leave the cells via the Golgi apparatus where the spike protein undergoes also proteolytic cleavage by furin. This process preferably occurs in virus producer cells with high TMPRSS2 expression, e.g., alveolar cells (**left side**). Alternatively, the virus can be taken up via clathrin coated pits into endosomes, followed by proteolytic cleavage by cathepsin L. The endosomal uptake is predominant in TMPRSS2-negative, but cathepsin L-rich cells, such as innate immune cells and endothelial cells (**right side**). The spike protein of the SARS-CoV-2 acts as a TLR4 agonist, resulting in dimerization of the TLR4/MD-2 complex triggering downstream signaling, e.g., activation of the NF-κB (p50/p65) pathway. The activation of NF-κB pathway triggers HIF-1α activation and expression of cytokines, such as TNFα, IL-1, IL-6, IL-12. Notably, HIF-1α and IL-12 have been found to enhance furin expression. In contrast to the highly effective TLR4/MD-2 dimerization and activation by the wild-type SARS-CoV-2 spike protein trimer, the amino acid substitutions in the Omicron spike protein decrease the potency of the spike protein for TLR4/MD-2 dimerization, leading to less NF-κB signaling and lower expression of cytokines. (**B**,**C**) The relative expression of TLR4 differs depending on the cell type, with high expression of TLR4/MD-2 on innate immune cells, (**C**) but only low levels expressed on alveolar lung cells (**B**). Figure adapted from [11].

**Table 1 ijms-25-05451-t001:** Summary of Top 10 Models HDOCK docking for wild type spike protein (PDB 6ZGG) and TLR4/MD2 (PBD 3FXI).

Rank	1	2	3	4	5	6	7	8	9	10
Docking Score	−347.62	−332.77	−313.50	−311.94	−310.11	−305.95	−299.03	−298.52	−291.15	−289.51
Confidence Score	0.9812	0.9748	0.9634	0.9623	0.9609	0.9517	0.9577	0.9512	0.9439	0.9421

**Table 2 ijms-25-05451-t002:** Summary of Top 10 Models HDOCK docking for Omicron spike protein (PDB 7ZL9) and TLR4/MD2 (PDB 3FXI).

Rank	1	2	3	4	5	6	7	8	9	10
Docking Score	−368.16	−337.33	−326.17	−321.62	−306.41	−305.44	−304.69	−304.24	−300.36	−296.86
Confidence Score	0.9874	0.9769	0.9713	0.9687	0.9580	0.9573	0.9566	0.9563	0.9529	0.9497

## Data Availability

Data is contained within the article and Appendix A.

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
