# Peer review of "In Silico Analyses Indicate a Lower Potency for Dimerization of TLR4/MD-2 as the Reason for the Lower Pathogenicity of Omicron Compared to Wild-Type Virus and Earlier SARS-CoV-2 Variants"

_ijms, 2024, doi:10.3390/ijms25105451_

Round 1

Reviewer 1 Report

Comments and Suggestions for Authors

SARS-CoV-2, an enveloped positive-sense, single-stranded RNA virus, attaches to classical receptor protein angiotensin-converting enzyme-related carboxypeptidase-2 (ACE-2) receptors on target cells via its spike (S) protein. In addition to ACE2, alternative cellular receptors have been identified as potential binding targets for the SARS-CoV-2 virus, including integrins, sialic acid, heparan sulfate, CD147, and neuropilin-1. Toll-like receptors (TLRs) play a crucial role in recognizing and eliminating pathogen-associated molecular patterns from viruses.

The author conducted in silico analyses indicating that the Omicron spike protein has reduced potency in inducing dimerization of TLR4/MD-2 compared to the wild-type virus, despite exhibiting comparable binding activity to TLR4. While the author extensively analyzed TLR4/MD-2 dimerization during spike protein interaction, there are several concerns that need to be addressed for the paper's acceptance. The following comments are provided:

  1. 1. Given the high expression of TLR2 and TLR4 in innate immune cells, it would be valuable to include binding predictions of TLR2 with the spike protein.

  2.  
  3. 2. Is there a possibility of TLR2 dimerization during interaction with the spike protein? This aspect warrants investigation.

  4.  
  5. 3.Although current in silico investigations suggest TLR4/MD-2 complex dimerization upon interaction with spike S1 and S2 subunits, it remains unknown whether this dimerization occurs in vitro cultured cells or in vivo. Additional data may be necessary to address this uncertainty.

  6.  
  7. 4. Are there any differences in TLR4/MD-2 dimerization between different variants of SARS-CoV-2? This comparison could provide insights into the virus's pathogenicity and immune evasion mechanisms.

  8.  
  9. 5. Given increasing reports of neurological symptoms associated with COVID-19, it would be beneficial for the author to explore TLR interactions with microglia-related receptors or cite relevant literature (e.g., ACS Pharmacol. Transl. Sci. 2023, 6, 1323−1339; Expert Rev Neurother. 2023 Jul-Dec;23(8):703-720) and summarize potential mechanisms of TLR interaction with the spike protein.

  10.  
  11. 6. Considering that antibodies are a popular and potent tool for combating SARS-CoV-2 infection, the author should discuss the potential of antibody-based therapies targeting TLR4/MD-2 dimerization. Relevant literature (e.g., ACS Pharmacol. Transl. Sci. 2023, 6, 925−942; Expert Opin Biol Ther. 2024 Mar;24(3):191-201) could be cited to support this discussion.

Incorporating these suggestions would enhance the comprehensiveness and relevance of the paper, addressing critical aspects of SARS-CoV-2 pathogenesis and potential therapeutic strategies.

Author Response

The author appreciates the comments by the reviewer.

All topics have been addressed in the attached response letter and in the new manuscript, all relevant changes are marked in yellow

Reviewer 2 Report

Comments and Suggestions for Authors

The topic is important and the manuscript provides a comprehensive analysis of the subject. However, some details and issues are required before publication, which are listed as follows.

1.     “SARS-CoV-2 virus” please change with “SARS-CoV-2”.

2.     Ensure that you provide clear and concise explanations of your methodology and findings. Consider discussing the limitations of your study and potential future directions for research. Addressing these limitations will add credibility.

3.     As new variants continue to evolve, the authors seem to ignore the importance of drug resistance in drug discovery.

4.     There is a lack of related literature citations. Please include relevant content in the relate sections. In lines 33-35, “This correlates with Omicron’s higher transmission rate within the population while causing less frequently acute respiratory distress syndrome (ARDS) and severe clinical symptoms of COVID-19. (DOI: 10.3390/nu15153443)”.; in lines 123-125, “There is increasing evidence that Omicron despite a significantly higher transmissibility and infectivity shows a significantly lower rate of severe clinical courses compared to previous VoCs, including the Delta variant, i.e. the last VoC before Omicron. (DOI: 10.1016/j.ejmech.2023.115503)”

5.     To provide more reliable evidence, it is necessary to further discuss the problems existing in the study.

Comments on the Quality of English Language

Minor editing of English language required

Author Response

The author appreciates the evaluation and the comments raised by the reviewer and has addressed all points in the new manuscript. All paragraphs and references to these points are marked in blue

  1. SARS-CoV-2 virus has been changed to SARS-CoV-2
  2. New Paragraphs describing methodology, explanation of findings, limitations of the present study and potential future directions have been included in the new manuscript, please refer to Discussion and Materials & Methods sections (marked in blue)
  3. The topic of development of drug resistance by the virus evolution has been addressed in the Discussion (marked blue)
  4. the indictaed refernces have been included in the manuscript as suggested by the reviewer (marked blue)
  5. Limitations have been addressed in a paragraph in the Discussion section. Furthermore, some additional data sets as requested by the second reviewer and which address some of the problems/limitations have been addressed in the new manuscript (marked in yellow, the response letter attached for information)

Round 2

Reviewer 1 Report

Comments and Suggestions for Authors

I have no further comments on this manuscript, and I would suggest to accept it.